# Exacerbation of Hangover Symptomology Significantly Corresponds with Heavy and Chronic Alcohol Drinking: A Pilot Study

**DOI:** 10.3390/jcm8111943

**Published:** 2019-11-12

**Authors:** Vatsalya Vatsalya, Hamza Z. Hassan, Maiying Kong, Bethany L. Stangl, Melanie L. Schwandt, Veronica Y. Schmidt-Teron, Joris C. Verster, Vijay A. Ramchandani, Craig J. McClain

**Affiliations:** 1Department of Medicine, University of Louisville, Louisville, KY 40202, USA; hamza.hassan@louisville.edu (H.Z.H.); craig.mcclain@louisville.edu (C.J.M.); 2Alcohol Research Center, University of Louisville, Louisville, KY 40202, USA; 3Hepatobiology & Toxicology Center, University of Louisville, Louisville, KY 40202, USA; 4National Institute on Alcohol Abuse and Alcoholism, NIH, Bethesda, MD 20892, USA; bethany.stangl@nih.gov (B.L.S.); melanies@mail.nih.gov (M.L.S.); vschmidt@uchc.edu (V.Y.S.-T.); vijayr@mail.nih.gov (V.A.R.); 5Robley Rex Louisville VAMC, Louisville, KY 40206, USA; 6Department of Bioinformatics and Biostatistics, SPHIS, University of Louisville, Louisville, KY 40202, USA; maiying.kong@louisville.edu; 7Division of Pharmacology, Utrecht Institute for Pharmaceutical Sciences (UIPS), Utrecht University, 3584 CG Utrecht, The Netherlands; J.C.Verster@uu.nl; 8Faculty of Veterinary Medicine, Institute for Risk Assessment Sciences (IRAS), Utrecht University, 3511 CM Utrecht, The Netherlands; 9Centre for Human Psychopharmacology, Swinburne University, Melbourne, VIC 3211, Australia; 10Department of Pharmacology & Toxicology, University of Louisville, Louisville, KY 40202, USA

**Keywords:** alcohol hangover scale (AHS), alcohol use disorders identification test (AUDIT), dependence symptoms of AUDIT (DS-AUDIT), hangover, heavy drinking

## Abstract

Alcohol hangover is a combination of mental, sympathetic, and physical symptoms experienced the day after a single period of heavy drinking, starting when blood alcohol concentration approaches zero. How individual measures/domains of hangover symptomology might differ with moderate to heavy alcohol consumption and how these symptoms correlate with the drinking markers is unclear. We investigated the amount/patterns of drinking and hangover symptomology by the categories of alcohol drinking. We studied males and females in three groups: 12 heavy drinkers (HD; >15 drinks/week, 34–63 years old (y.o.)); 17 moderate drinkers (MD; 5–14 drinks/week, 21–30 y.o.); and 12 healthy controls (social/light drinkers, SD; <5 drinks/week, 25–54 y.o.). Demographics, drinking measures (Timeline followback past 90 days (TLFB90), Alcohol Use Disorders Identification Test (AUDIT)), and alcohol hangover scale (AHS) were analyzed. Average drinks/day was 5.1-times greater in HD compared to MD. Average AHS score showed moderate incapacity, and individual measures and domains of the AHS were significantly elevated in HD compared to MD. Symptoms of three domains of the AHS (mental, gastrointestinal, and sympathetic) showed domain-specific significant increase in HD. A domain-specific relation was present between AUDIT and specific measures of AHS scores in HD, specifically with the dependence symptoms. Exacerbation in hangover symptomology could be a marker of more severe alcohol use disorder.

## 1. Introduction

Excessive alcohol consumption is a leading cause of preventable mortality in the United States [1]. However, alcohol consumption continues to steadily rise in the United States; 2014 year estimates (52.7% of people aged 12 or older drank in the past month of reporting) were higher than the estimates in most years between 2002 and 2008 [2]. The most frequently reported consequence of excessive alcohol consumption is experiencing a hangover. The alcohol hangover refers to the combination of mental and physical symptoms, experienced the day after a single episode of heavy drinking, starting when blood alcohol concentration approaches zero [3]. The number of people experiencing alcohol hangover is high, with studies reporting 78% and higher prevalence of alcohol induced hangover [4,5] including social/light, moderate, and heavy drinkers. Hangover symptoms can be severe enough to impair daily routine, reduce productivity, and cause other associated complications of alcohol consumption [6]. Thus, understanding these symptoms and how they relate to the severity of alcohol abuse is important. 

Adverse effects of excess alcohol intake manifest acutely as hangover symptoms. “Hangover” consists of a wide array of mental, physical, and neuropsychological (including sympathetic) symptoms that occur usually within hours of alcohol consumption and are recorded through the following day starting when blood alcohol concentration approaches zero [7,8,9]. Some of the common symptoms are fatigue, headaches, nausea, sleepiness, shakiness, weakness, excessive thirst and dry mouth, mood disturbances, and apathy [10,11,12].

Although hangover has been linked to heavy alcohol consumption and binge drinking, it can also be seen in individuals with moderate alcohol drinking [13,14]. This observation illustrates the gaps in our knowledge of how hangover symptoms manifest differently in moderate and heavy drinkers. Further, we do not know how hangover symptomology changes with the increased/altered drinking volume/patterns. Recent drinking history (Timeline follow-back, TLFB), and Alcohol Use Disorders Identification Test (AUDIT) are validated measures for assessing drinking patterns and quantity. In recent investigations, AUDIT and TLFB were used to study hangover symptoms [12,15] and they showed a close association with the hangover spectrum. Such associations have not been investigated in individuals with a high level of heavy drinking (≥15 drinks/per day), nor have there been studies of how hangover symptoms change with increased levels of alcohol drinking and altered patterns of consumption.

The primary objective of this pilot study was to identify the domains and individual measures of hangover symptoms that are different in individuals who consume alcohol in a moderate fashion and those who are heavy drinkers. Another aim of this study was to identify the associations of hangover symptoms and heavy drinking markers (derived from drinking history assessments) in heavy drinkers. We also included some of the potential modifiers (sex, family history, comorbid conditions) contributing to the changes in drinking patterns and hangover symptoms and assessed their interactions.

## 2. Study Participants and Methods

### 2.1. Patient Recruitment

The specific investigations reported here were conducted under two different larger protocols, one approved by the Combined Neuroscience Institutional Review Board at the NIH and other approved by the University of Louisville IRB. IRBs of both institutions approved the study once it was concluded that the study objectives met the ethical standards/regulations. The study at the NIH was indexed at ClinicalTrials.gov with identifier #NCT00713492, and the studies carried out at the University of Louisville were indexed at ClinicalTrials.gov, with the identifier numbers: #NCT01922895; #NCT01809132. Study participants who met diagnosis criteria for alcohol use disorder (AUD) and alcoholic liver disease (ALD) were approached at the outpatient and inpatient settings of the University of Louisville. Moderate drinkers and healthy volunteers were approached by advertisement using flyers, newspaper column, and word of mouth at both sites. All participants meeting eligibility criteria were enrolled after providing written informed consent. All participants’ data were available securely to the PIs only. De-identified data were used to perform statistical analyses. Heavy drinkers and social/light drinkers were recruited at the University of Louisville; and moderate drinkers were recruited at NIH using cohort specific eligibility criteria as mentioned further.

A total of forty-one male and female individuals aged 21 to 64 years participated in this study. Study participants were classified based on their alcohol consumption pattern as follows: heavy drinkers ((HD), *n* = 12, aged 34–64 years); moderate drinkers ((MD), *n* = 17, aged 21–30 years); and social/light drinkers (healthy volunteers (SD); *n* = 12, aged 25–54 years). In our study, heavy drinkers (NIAAA guideline: ≥15 drinks/week for females; ≥20 drinks/week for males) drank around 15 drinks per day on an average (>90 drinks/week); all participants also met criteria for alcohol use disorder diagnosis according to DSM 5 criteria. Moderate drinkers drank in between 5–14 alcoholic drinks per week. Social/light drinkers (SD) drank four or fewer drinks on an average per week; they were not abstinent to alcohol drinking and were actively drinking at the time of screening. Inclusion criteria for heavy drinkers were that the females drank 15 or more drinks/week, and males drank 20 or more drinks/week meeting the NIAAA guidelines. Inclusion criteria for heavy drinkers also included being age 21 years or older with reported heavy drinking for at least the past six months. Inclusion criteria for the social/light drinkers (SD) in this study were: (a) age 21 years or older, (b) without any reported heavy or moderate drinking for at least the past six months, (c) normal comprehensive metabolic panel (normal liver and kidney panel, specifically serum albumin, total bilirubin, aspartate aminotransferase (AST), and alanine aminotransferase (ALT), and (d) no ongoing inflammation/infection or occurrence in the last three months. Exclusion criteria for heavy drinkers were: (a) unwilling or unable to provide informed consent, (b) significant comorbid conditions (heart, kidney, lung, neurological or psychiatric illnesses, sepsis) or active drug abuse, (c) pregnant or lactating women, (d) other known liver disease (except alcoholic liver disease—ALD), and/or (e) prisoners or other vulnerable subjects. Exclusion criteria for social/light drinkers included clinical diagnosis of any kind of liver disease and alcohol consumption meeting heavy or moderate drinking classification plus the four criteria above for heavy drinkers. Moderate drinkers (MD) had the same inclusion and exclusion as of social/light drinkers apart from the drinking profile (clause b). All study participants had a complete history and physical examination and laboratory evaluation upon study enrollment.

### 2.2. Study Paradigm

This investigation was a single time point assessment of AUD patients and healthy control participants. All participating individuals were consented for this study prior to collection of data and bodily samples. We collected demographic data, clinical data, medical history, and biochemical measures of liver injury and dysfunction. We also collected drinking history information (using timeline followback (TLFB), and Alcohol Use Disorders Identification Test (AUDIT)) from heavy drinkers and TLFB from moderate and social/light drinkers. Hangover symptomology was assessed using the Alcohol Hangover Scale (AHS). Participants reported on hangover after the last alcohol consumption until 10:00 am of the following day. The following drinking markers were calculated from the TLFB assessment for the past 90 days: total drinks past 90 days (TD90), number of drinking days past 90 days (NDD90), average drinks per drinking day past 90 Days (AvgDPD90), and heavy drinking days past 90 Days (HDD90) [16]. We used the total AUDIT score [17] and scores for its individual domains: hazardous alcohol use (frequency of drinking, typical quantity, and frequency of heavy drinking (HzAU)), dependence symptoms (impaired control over drinking, increased salience of drinking, and morning drinking (DS)) and harmful alcohol use (guilt after drinking, blackouts, alcohol-related injuries (HAU)). AHS is one of the standardized symptom scales used to quantify intensity of hangover symptoms based on the two publications we used as our assessment for hangover symptoms [18,19]. We used the guideline for AHS data based on our previous publication [12]. Symptoms measured by the AHS included mental domain (“hangover”, “dizziness”, and “craving”), physical domain measures (“thirsty”, “tired”, and “headache”), gastrointestinal domain symptoms (“nausea”, “loss of appetite”, and “stomach ache”) and sympathetic (“heart-racing”) that were analyzed individually and domain-wise. Participants rated the hangover symptoms they experienced until 10 am of the following morning after alcohol consumption on a scale from 0–7; as described in a previous publication from our group [12]. Categories of severity for hangover scale are n = 0 as “none”, n = 1–3 as “mild”, n = 4–6 as “moderate”, and n = 7 as “incapacitated”. 

Family history of alcoholism (FHA) assessment was performed on all the subjects [20]. Subjects were asked for a family history of alcohol use disorders. A positive history (FHP) meant having at least one or more first degree relatives with a history of alcohol use disorder. A negative family history (FHN) was confirmed as being without any first- or second-degree relatives with alcohol use disorder.

Liver injury markers alanine transaminase (ALT), aspartate aminotransferase (AST), the ratio of AST:ALT, albumin, and total bilirubin were examined. All these measures were analyzed under comprehensive metabolic panel of the standard of care order. Laboratory testing were performed by the hospital laboratory at both the institutions. 

### 2.3. Analysis

Demographics, drinking history and hangover data were analyzed using one-way analysis of variance (ANOVA) along with post-hoc t-tests for group comparisons. Group means and standard deviations were tabulated. Assessment of the association of hangover symptoms (itemized hangover measures, their domains, and average hangover score (avgAHS)) with AUDIT scores, TLFB measures, and family history of alcoholism were evaluated using multiple linear regression model, where sex and age were selected as co-variables; and liver function markers using liver panel tests (serum albumin and serum total bilirubin) were incorporated as modifying factors in multivariate linear regression models. SPSS 25.0 (IBM, Chicago IL, USA), GraphPad Prism 7.0 (GraphPad Software, San Diego CA, USA), and Microsoft Excel 2016 (Microsoft Corp., Redmond WA, USA) software were used for data processing, statistical analyses, and plot/figure development. Descriptive data are presented as mean ± standard deviation (M ± SD). Effect size is shown in Figures 2–4 as a model fit (goodness-of-fit) of the relation (adjusted R^2^). Statistical significance was set at *p* < 0.05.

## 3. Results

### 3.1. Demographics and Drinking Profile

More females than males (almost double) were enrolled in both the moderate drinkers (MD) group (*n* = 11 and *n* = 6, respectively) and the social/light drinkers (SD) groups (*n* = 7 and *n* = 5, respectively). However, only approximately 1/4 of the heavy drinkers (HD) were female (*n* = 3 of 12; see Table 1). HD individuals had significantly higher (*p* = 0.023) BMI and were significantly (*p* ≤ 0.001) older. In our study, most of the HD individuals were family history positive for alcohol use disorder. The likelihood ratio for positive family history of alcohol use disorder in HD group was considerably higher (+30%, 6.971) at *p* = 0.008 (2-sided) compared to MD. By race, SD group had only one subject as Asian, and two subjects as African American. One subject was African American in the HD group. One individual reported multi-race background in the MD group. All study subjects but one in MD group by ethnicity were Non-Hispanic or Latino. As anticipated, all TLFB90 markers were significantly higher in HD group compared to SD or MD groups. The MD group showed numerically higher levels of all the TLFB90 measures compared to SD; however, only AvgDPD90 scores were statistically significant (Table 1). The mean AUDIT score in the HD group was >20, thus meeting criteria for diagnosis of alcohol dependence [17]. Ten of the 12 HD subjects had an AUDIT score >20. We did not evaluate sex differences since there were only three female heavy drinkers (Table 1).

### 3.2. Assessment of Hangover Symptoms

There was a stepwise pattern to the occurrence of hangover symptomology wherein almost all of the hangover measures increased numerically from the social to moderate drinker groups and were the highest in heavy drinkers (Table 2). We found a large variability in some of the measures (as observed by the standard deviations) (Figure 1). This could be due to the individual variability in how the individual measures and domains of hangover manifest.

### 3.3. AUDIT Domains, and Association of Drinking Markers in Heavy Drinkers

The HzAU domain of the AUDIT was significantly higher compared to DS and HAU domains in HD group (Figure 2a). Neither the DS nor the HAU domain exerted as great an effect as the HzAU (Figure 2a). The AUDIT score and the recent heavy drinking marker, HDD90, showed a significant association in the HD group (Figure 2b). Among the domains of AUDIT, only HzAU domain was significantly associated with HDD90 marker at moderate effects in the HD group (Figure 2c), thus suggesting overall AUDIT vs. heavy drinking relation was primarily due to the hazardous domain of AUDIT. 

### 3.4. Internal Consistency of Hangover Measures in Heavy Drinkers

Mental domain measures of “hangover”, “craving”, and “dizziness” together showed very strong main effects, unadjusted R^2^ = 0.707 at high significance *p* = 0.009. Physical domain measures of “headache”, “thirsty”, and “tired” together showed the strongest main effects, unadjusted R^2^ = 0.862 at very high significance *p* ≤ 0.001. Gastrointestinal domain measures of “stomachache”, “nausea”, and “loss of appetite” together also showed strong main effects, unadjusted R^2^ = 0.842 at high significance *p* = 0.001. “Stomachache” and “nausea” measures were associated at high effects adjusted R^2^ = 0.642 and high significance p = 0.001. No other measures within any of the domains showed any such association. 

### 3.5. Association of AUDIT and Hangover Measures in Heavy Drinkers

In the HD group, the AUDIT showed a significant association with the average hangover scores (Figure 3a) and with three hangover measures: “heart-racing” (Figure 3b), “craving” (Figure 3c), and “thirsty” (Figure 3d). The AUDIT score showed significant high main effects with the symptoms of mental domain (“hangover”, “dizziness”, and “craving”) (unadjusted R^2^ = 0.727, *p* ≤ 0.001), as well as physical domain (“thirsty”, “tired”, and “headache” together) (unadjusted R^2^ = 0.778, *p* ≤ 0.001) in this group. Gastrointestinal domain symptoms (“nausea”, “loss of appetite”, and “stomachache”) showed significant, though moderate, main effects with AUDIT scores (unadjusted R^2^ = 0.685, *p* ≤ 0.001) in these heavy drinkers.

### 3.6. Role of AUDIT and Heavy Drinking TLFB Markers on Hangover Symptoms

There was no direct association of hangover measures and timeline followback markers (data not plotted). However, specific TLFB drinking markers and AUDIT scores together showed the augmented association with hangover measures, and that effect was higher than those exhibited by the univariate associations of AUDIT and specific hangover measures (Figure 3a,c). The average hangover score and AUDIT score showed a strong association (adjusted R^2^ = 0.578) at high significance, *p* = 0.021, when adjusted for NDD90. The “craving” measure and the AUDIT showed an association (adjusted R^2^ = 0.462) at a high significance (*p* = 0.025), when adjusted for HDD90.

We further evaluated the association of hangover measures and domains of AUDIT to identify domain-specific relations in heavy drinkers. The hangover measure, “heart-racing”, was closely associated with the combined effect of hazardous alcohol use (HzAU) and the “dependence symptoms” (DS) domains of the AUDIT (Figure 4a,d). Notably, the association of “heart-racing” values and DS scores was of very high effect (Figure 4d). DS scores also correlated with average hangover scores (Figure 4b), for “craving” (Figure 4c), and for “stomachache” (Figure 4e) in heavy drinkers. The association of DS and craving was likely significant (Figure 4c); when this association was tested with HDD90 as a covariate, both the significance (*p* = 0.025) and effect size (Adjusted R^2^ = 0.462) showed substantial augmentation. The harmful alcohol use (HAU) domain was significantly associated with only one hangover measure, “thirsty”, in heavy drinkers (Figure 4f). Domain specific responses of the AUDIT showed close associations with specific hangover symptoms.

### 3.7. Role of Liver Dysfunction on Hangover Symptoms

Impaired liver health, characterized by serum total bilirubin and serum albumin (Table 3), along with drinking profile (using AUDIT) was predictive of the average hangover score (Figure 5). However, we did not find major effects of liver injury markers (ALT, AST, and AST:ALT ratio) in combination with AUDIT on hangover severity.

In heavy drinking AUD patients, more severe form of ALD was a comorbid condition (with a diagnosis of alcoholic hepatitis) (Table 3). Heavy drinkers with comorbid ALD showed signs of both clinical range of liver injury and function markers. AUDIT scores showed significant association with average hangover score and other hangover symptoms (Figure 3), and thus, we investigated if there is any mediating role of liver function/ injury in the hangover symptomology (Figure 5).

One of the hypotheses for our observation is that impaired liver function contributes to suboptimal metabolism of alcohol and alcohol metabolites. This could lead to both higher blood alcohol concentrations and longer durations in the system. Aldehyde, a metabolite of alcohol metabolism primarily in liver, has toxic effects on the system that lead to pathological (oxidative stress), and this may be related to psychiatric manifestations (withdrawal/hangover). We found that higher hangover scores are associated with greater effects by heavy drinking (AUDIT) mediated by worsening indications of liver function (Figure 5). Further investigation on the roles of alcohol dehydrogenase and aldehyde dehydrogenase with respect to hangover symptomology could help define this relation. 

## 4. Discussion

All heavy drinkers in this study reported hangover symptoms, whereas only some moderate drinker subjects reported hangover symptoms. The average hangover score in heavy drinkers was >3 of a possible 7, which was, by definition, a moderate level of incapacity. Three measures of hangover, “thirsty”, “craving”, and “loss of appetite”, were numerically well into the moderate range of hangover intensity. Mental, gastrointestinal and sympathetic domains of the hangover scale were significantly higher in heavy drinkers compared to the moderate drinkers. Almost all of the HD patients reported a positive family history of alcoholism in our study. The AUDIT score has been shown to have a significant association with future alcohol use disorder, with up to 61% of higher AUDIT scores linked to alcohol use disorder in comparison to 10% with lower scores [21]. In our study, we found the majority of the heavy drinkers scored very high (>20) on the AUDIT scale. We found a close association of heavy drinking days past 90 days (HDD90, a component of TLFB90) with the AUDIT score and with its hazardous domain in particular. In our previous findings, we found a close association of recent drinking and hangover in moderate drinkers [12]. In the heavy drinkers, hangover measures showed a close association with AUDIT and its domains. Recent studies have linked higher hangover symptoms to a higher risk of alcohol use disorder and delirium tremens [22]. However, there is a significant gap in the understanding of the intensity of hangover symptoms and their association with patterns of heavy alcohol drinking. Our previous study reported a mild level of hangover symptoms in moderate drinkers [12]. In this study, we found that the heavy drinkers, who also exhibited remarkably heavier alcohol intake, showed a moderate level of incapacity in overall hangover assessment. 

AUDIT scores (and specifically hazardous and dependence domains of the AUDIT scale) of heavy drinkers were most closely associated with the “heart-racing” measure of hangover. “Heart-racing” is a measure of the sympathetic domain of hangover, and manifestations of sympathetic hyperactivity could be related to vasoconstriction and tachycardia [23]. Stimulant and sedative use can potentiate sympathetic nervous system hyperactivity that could be related to elevated “heart-racing” and may result in severe withdrawal symptoms in alcohol use disorder. 

In our study, the AUDIT score and its harmful domain showed a close association with the physical (“thirsty”) domain. This association is consistent with a previous publication stating that hangover could be an indicator of risk for physical dependence [24]. Likewise, the AUDIT score and its dependence domain showed a significant association with the mental domain “craving” measure. Subsequent drinking to relieve hangover could lead to heavier alcohol consumption and higher AUD symptoms [25]. Further, craving and its association with the dependence domain of AUDIT might influence subsequent drinks and their timing, as has been reported previously [26], also suggesting increased predisposition for alcohol use disorder (AUD). Hangover symptomology is a direct and immediate adverse effect of alcohol intake, which we have now assessed in all categories of drinking (social, moderate, and heavy) in this pilot study. Importantly, we have shown a stepwise relation between levels of drinking and symptomology of hangover. However, in the alcohol use disorder spectrum, much emphasis is given on dependence and withdrawal categories of diagnosis. In heavy drinkers, these adverse symptoms of alcohol-associated hangover are worse than in moderate or social/light drinkers. Furthermore, the dependence domain of the AUDIT and specific domains of the hangover scale correlate highly in heavy drinkers. Thus, these findings provide the basis for including alcohol-associated adverse effects scales, such as the AHS, in the diagnosis of alcohol use disorder (detailing item 3 of the “DSM-5” concerning the aftereffects: https://pubs.niaaa.nih.gov/publications/dsmfactsheet/DSMfact.pdf). Validation of these findings and their effects from a large study cohort of heavy and moderate drinkers may elucidate the role of specific adverse effects of alcohol intake in the development and progression of alcohol dependence. 

Most studies report that males consumed a higher amount of alcohol in comparison to females; thus, males are likely to show higher incapacity in the hangover symptoms scale [27]. In our study, 3/4 of the participants in HD group were males, and they exhibited numerically higher severity both in alcohol consumption and hangover symptoms. We examined other factors that could be associated with higher alcohol consumption. Positive family history of alcoholism was a significant factor associated with heavy alcohol consumption and higher hangover symptom scores, as reported previously [28,29,30]. Our study was consistent with those findings.

There were limitations in this study. This study was performed as a proof of principle design to address the primary aims. Given the results showing considerable effect sizes, this study supports the utility of these outcomes in larger populations. We acknowledge that these initial results would benefit from further testing with a larger sample size, and we are pursuing such study presently. We did not collect the length of time for each of the hangover symptoms. This should likely be used as another parameter in the ongoing and future studies. There were few females in the heavy drinking group, therefore we could not determine any sex differences. Variability in ethnicity was limited, most of the participants were Caucasian, and few were Hispanic-Latino; thus, any meaningful race/ethnicity-based comparisons is not within the scope of this study. Gamma-glutamyl transferase (GGT), a marker of liver injury was not tested in this study. Our focus in this study was on the heavy drinkers; 11 of 12 participants were positive for family history of alcoholism. Importantly, the sample size for each of the subgroups is not large, and thus, we focused on the effect sizes for interpreting the results. There are more factors that can modify/potentiate hangover symptoms, and assessing those was not in the scope of this study. For example, dehydration and electrolyte imbalances can worsen significantly under the effects of alcohol, and we did not study either of these factors. Nor did we evaluate acute withdrawal symptoms (Clinical Institute Withdrawal Assessment for Alcohol Scale-revised (CIWA-Ar)) in heavy drinkers for comparisons with the AHS, the AUDIT, or the TLFB measures. This is a future direction. 

We found an interesting association between liver function markers (albumin and total bilirubin) and higher hangover scale results. One of the hypotheses for our observation is that impaired liver function contributes to slower metabolism of alcohol and alcohol metabolites. This could lead to both higher and more persistent blood alcohol/metabolite concentrations. We did not find any direct (or modifying) association of liver injury and higher hangover symptoms. This could inform that liver function (not liver injury) might be a better predictor of alcohol associated adverse effects. Further study could identify an altered mechanistic response that causes delayed and slower alcohol metabolism that would be consequential in higher and prolonged levels of alcohol consumption. 

The presence and severity of hangover symptoms varied substantially across the three drinking groups. Hangover in heavy drinkers was debilitating, while in moderate drinkers, it was reported only as uncomfortable. Domains of hangover symptoms corresponded with the specific markers/domains of heavy alcohol drinking in heavy drinkers. Specific assessment of drinking measures should be used for assessing hangover symptomology in different drinking groups; AUDIT seems to better correlate with hangover measures in heavy drinkers. Elevated domains of symptoms assessed by the AUDIT in heavy drinkers, and their relevance with domains of the AHS, support the potential use of the AHS in determining severity of alcohol use disorder. Our findings point to the need for further studies on larger groups of drinkers with regard to patterns of alcohol consumption, levels of alcohol use, and adverse symptomology. Alcohol associated hangover, adverse effects, and liver injury are most common immediate manifestations that are observed together very frequently. AUD characterization based on DSM-5 criteria addresses some of the hangover symptomology in the diagnosis; however, it does not specify the integrated role of the aforementioned measures. Available treatment options for AUD are also limited. Shared pathology of AUD and ALD require focused treatment for both AUD and ALD. Disulfiram and Acamprosate can be useful in AUD patients who are abstinent at the time of treatment; and Naltrexone cannot be prescribed in AUD patients with liver injury (black box warning identifies adverse interaction with liver health). Thus, intervention for AUD patients with liver injury who are actively drinking heavily is limited and new drugs are under investigation, for example, Varenicline tartrate, which does not interfere with liver [31]. Specifically, larger groups are needed to evaluate any potential sex differences; we have reported about the differences in liver injury and drinking patterns in AUD females in a larger study population previously [32]. 

## 5. Conclusions

Alcohol hangover symptoms increase with corresponding increases in the level of alcohol consumption. Specific domains of hangover symptomology are exacerbated with heavy drinking. Symptoms of hangover are associated with long term assessment of heavy drinking rather than the recent drinking history. AUDIT (Alcohol Use Disorders Identification Test) and its hazardous domain are associated with the heavy drinking marker, HDD90 (heavy drinking days in the past 90 days). Dependence scores of AUDIT scale were closely associated with the presentation of adverse after-effects of heavy drinking as characterized by the hangover scale (chiefly in the mental, sympathetic, and gastrointestinal domains). Large population studies are needed to confirm and more precisely illustrate the findings in this study and would also allow for analysis of differences between sexes.

## Figures and Tables

**Figure 1 jcm-08-01943-f001:**
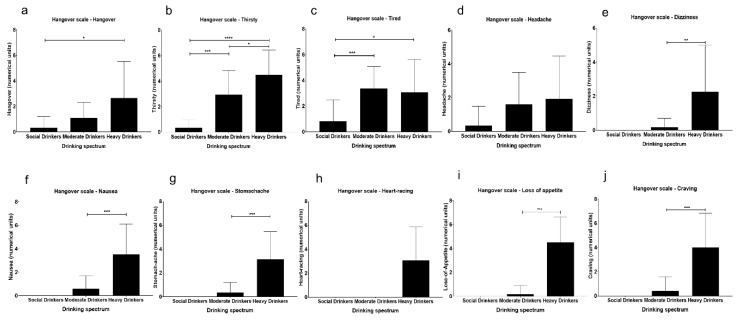
Levels of hangover measures in social/light drinkers (SD), moderate drinkers (MD), and heavy drinkers (HD). (**a**): Hangover measure “Hangover”. (**b**): Hangover measure “Thirsty”. (**c**): Hangover measure “Tired”. (**d**): Hangover measure “Headache”. (**e**): Hangover measure “Dizziness”. (**f**): Hangover measure “Nausea”. (**g**): Hangover measure “Stomachache”. (**h**): Hangover measure “Heart-racing” (**i**): Hangover measure “Loss of appetite”. (**j**): “Craving”, “loss of appetite”, and “thirsty” measures showed moderate level of hangover severity in heavy drinkers (Table 2). Data are presented as mean ± standard deviation. Statistical significance was set at *p* ≤ 0.05. Statistical significance is not described in comparisons when the SD group individuals reported “zero”, as in some measures.

**Figure 2 jcm-08-01943-f002:**
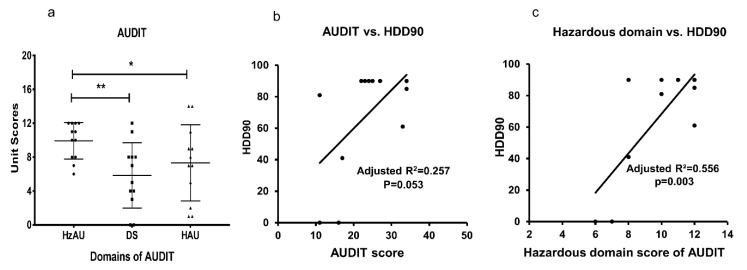
AUDIT and its association with timeline followback measures in heavy drinkers (HD). (**a**) Presentation of hazardous, dependency, and harmful domains that constitute AUDIT. (**b**) AUDIT score and heavy drinking days past 90 days (HDD90) drinking marker. (**c**) Scores of “hazardous-domain” of AUDIT and HDD90 (There are two points each for 11 and 12 [hazardous domain] that have corresponding 90 units of HDD90). Statistical significance was set at *p* ≤ 0.05.

**Figure 3 jcm-08-01943-f003:**
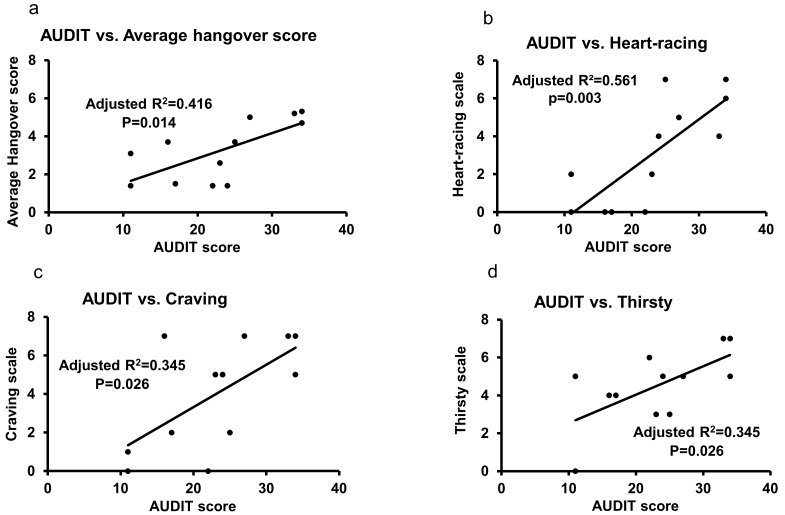
Association of AUDIT and hangover measures in heavy drinkers. (**a**) AUDIT and “Average hangover score” (AvgAHS). (**b**) AUDIT and “Heart-racing” (sympathetic domain) measure. (**c**) AUDIT and “Craving” (mental domain) measure. (**d**) AUDIT and “Thirsty” (physical domain) measure. Statistical significance was set at *p* ≤ 0.05.

**Figure 4 jcm-08-01943-f004:**
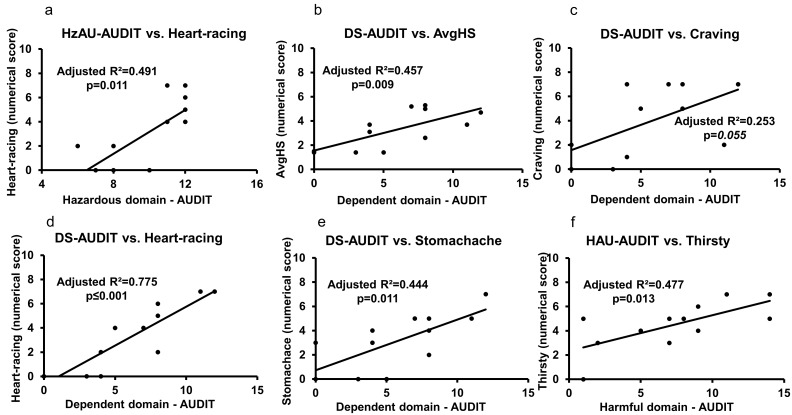
Association of AUDIT domains (Hazardous alcohol use (HzAU), dependence score (DS), and harmful alcohol use (HAU)) and individual hangover measures in heavy drinkers. (**a**) Association of HzAU and “Heart-racing” measure. (**b**) Association of DS and “Average hangover score” (AvgAHS) measure. (**c**) Association of DS and “Craving” measure. (**d**) Association of DS and “Heart-racing” measure. (**e**) Association of DS and “stomachache” measure. (**f**) Association of HAU and “Thirsty” measure. Statistical significance was set at *p* ≤ 0.05.

**Figure 5 jcm-08-01943-f005:**
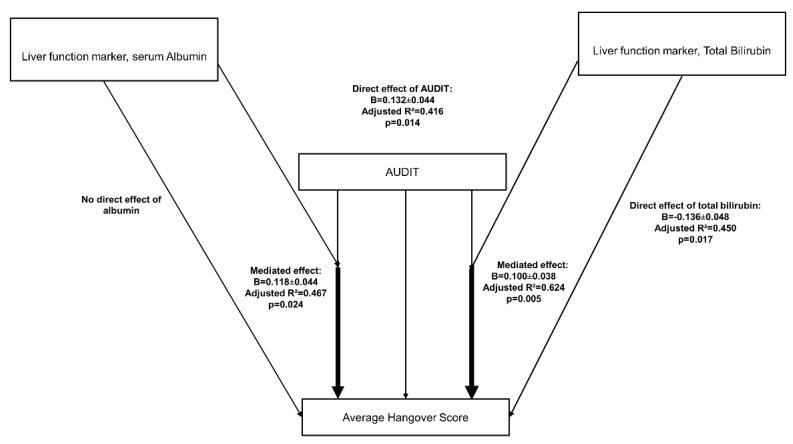
Schema of the effects of liver status (impaired liver function) and drinking severity on hangover symptomology. Liver function markers serum albumin and serum total bilirubin showed potential mediating roles in the effects of AUDIT on hangover symptoms.

**Table 1 jcm-08-01943-t001:** Demographics, family history of alcoholism, and drinking measures.

Measures	Heavy Drinkers (*n* = 12)	Moderate Drinkers (*n* = 17)	Social Drinkers (*n* = 12)	Heavy vs. Moderate Drinking Group Significance
**Demographics and Family History of Alcoholism**				
Sex (M or F)	*n*(M) = 9, *n*(F) = 3	*n*(M) = 6, *n*(F) = 11	*n*(M) = 5, *n*(F) = 7	NA
Age (years.) ^a,b^	49.8 ± 9.8	25.1± 3.1	31.6 ± 10.6	≤0.001
BMI ^a^	30.6 ± 7.6	25.1 ± 4.7	25.0 ± 3.1	0.023
FHA	FHP, *n* = 11; FHN, *n* = 1	FHP, *n* = 8; FHN, *n* = 9	FHP, *n* = 5; FHN, *n* = 7	NA
**Heavy Drinking Markers**				
TD90 ^a,b^	1153.92 ± 765.56	74.41 ± 31.39	23.67 ± 18.14	≤0.001
HDD90 ^a^	67.33 ± 34.88	4.82 ± 9.47	0.58 ± 1.38	≤0.001
NDD90 ^a^	72.5 ± 24.4	31.47 ± 16.62	18.25 ± 18.70	≤0.001
AvgDPD90 ^a,b^	13.57 ± 8.1	2.5 ± 0.99	1.4 ± 0.72	≤0.001
**AUDIT Scores and Its Domains**				
AUDIT	23.08 ± 8.19	NC	NC	NA
AUDIT>20	10/12	NC	NC	NA
HzAU	9.92 ± 2.15	NC	NC	NA
DS	5.83 ± 3.86	NC	NC	NA
HAU	7.33 ± 4.5	NC	NC	NA

Abbreviations; M: male; F: female; BMI: body mass index; FHA: Family history of alcoholism; FHP: family history positive; FHN: family history negative; TD90: total drinks past 90 days; HDD90: heavy drinking days past 90 days; NDD90: number of drinking days past 90 days; AvDPD90: average drinks per drinking day past 90 days; AUDIT: alcohol use disorder identification test; DS: dependence symptoms; HAU: harmful alcohol use; HzAU: hazardous alcohol use. NA: not applicable, NC: not collected. a: between group comparisons of heavy and social/light drinkers; all comparisons found to have *p* ≤ 0.001. b: between group comparison of moderate and social drinkers (age: *p* = 0.023; BMI: not significant; TD90: *p* ≤ 0.001; HDD90: not significant; AvgDPD90: *p* = 0.002; NDD90: *p* = 0.055).

**Table 2 jcm-08-01943-t002:** Scores of hangover symptoms in social, moderate, and heavy drinkers. Effects of difference between moderate and heavy drinkers.

Symptoms	Social Drinkers (SD) *n* = 12	Moderate Drinkers (MD) *n* = 17	Heavy Drinkers (HD) *n* = 12	MD vs. HD Effects (Adjusted R^2^)
Hangover ^a^	0.33 ± 0.89	1.12 ± 1.22	2.67 ± 2.87	*p* = *0.056*; R^2^ = 0.129
Thirsty ^a,b^	0.33 ± 0.65	2.94 ± 1.89	4.50 ± 1.93	*p* = 0.039; R^2^ = 0.149
Tired ^a,b^	0.83 ± 1.64	3.35 ± 1.73	3.08 ± 2.54	*p* = 0.736; R^2^ = 0.004
Headache	0.33 ± 1.16	1.59 ± 1.91	1.92 ± 2.54	*p* = 0.693; R^2^ = 0.006
Dizziness ^a^	0.00	0.18 ± 0.53	2.25 ± 0.53	*p* = 0.005; R^2^ = 0.259
Nausea ^a^	0.00	0.59 ± 1.12	3.50 ± 2.61	*p* ≤ 0.001; R^2^ = 0.214
Stomachache ^a^	0.00	0.35 ± 0.86	3.17 ± 2.29	*p* ≤ 0.001; R^2^ = 0.445
Heart-racing ^a^	0.00	0.00	3.08 ± 2.78	*p* ≤ 0.001; R^2^ = 0.441
Loss of Appetite ^a^	0.00	0.18 ± 0.73	4.50 ± 2.15	*p* ≤ 0.001; R^2^ = 0.689
Craving ^a^	0.00	0.41 ± 1.18	4.0 ± 2.83	*p* ≤ 0.001; R^2^ = 0.451
AvgHS ^a,b^	0.18 ± 0.41	1.14 ± 0.64	3.3 ± 1.58	*p* ≤ 0.001; R^2^ = 0.489

Abbreviations; SD: social drinkers (or light drinkers); MD: moderate drinkers; HD: heavy drinkers; AvgAHS: average hangover score. a significant statistical difference in hangover symptoms between social and heavy drinkers. b significant statistical difference in hangover symptoms between social and moderate drinkers. In italics: not significant.

**Table 3 jcm-08-01943-t003:** Presentation of candidate liver panel markers.

Measures	Heavy Drinkers (*n* = 12)	Moderate Drinkers (*n* = 17)	Social Drinkers (*n* = 12)	Heavy vs. Moderate Drinking Group Significance
ALT (U/L)	73.83 ± 61.00	18.82 ± 7.95	28.5 ± 28.77	0.001
AST (U/L)	174.75 ± 82.11	24.53 ± 6.27	28.7 ± 13.92	≤0.001
AST:ALT ratio	3.53 ± 2.56	1.39 ± 0.48	1.03 ± 0.56	0.002
Total Bilirubin (μmol/L)	9.72 ± 7.77	0.55 ± 0.35	0.610 ± 0.31	≤0.001
Albumin (g/dL)	2.66 ± 0.29	4.17 ± 0.26	4.08 ± 0.24	≤0.001

Abbreviations; ALT: alanine aminotransferase, AST: aspartate aminotransferase. All markers in social and moderate drinkers were not clinically significant.

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
