# Peer review of "Exacerbation of Hangover Symptomology Significantly Corresponds with Heavy and Chronic Alcohol Drinking: A Pilot Study"

_jcm, 2019, doi:10.3390/jcm8111943_

Round 1

Reviewer 1 Report

Thank you for allowing me to read your revised paper and addressing comments made previously.  I have some further comments if I may:

1) It is still unclear to me how the participants were recruited? 

2) While the small sample size used in the study has been acknowledged more directly now and discussed as a limitation, the study has also now been referred to  as a proof of principle study.  In my view, the 'exploratory'/ early stage nature of the study ought to now be reflected in the title and so the title of the paper ought to be amended.  In addition, there are still some statements in the paper that require to be re-worded, eg on p12 line 361/362 it is stated that " We acknowledge that expanding this study in a larger population would further support our results, and we are pursuing this presently".   However, given that it is not known that expanding the study would further support the results then a degree of caution ought to be exercised here such that the statement might say, for example, "We acknowledge that these initial results would benefit from further testing with a larger sample size, and we are pursuing this presently"

3) There are still a number of typographical errors/language issues evident in the paper that would benefit from re-checking, eg:

p2, lines 80/81 - Combined Neuroscience Institutional Review Board at the NIH, and the other approved by the University of Louisville IRB. IRBs of both the Institutions approved the study once concluded, meeting the ethical standards/regulations.

p3, line 98 - however, they were not abstinent (to alcohol drinking) and were actively drinking at the time of screening.

Page 9. line 276 - there seems to be a word missing - In heavy drinking AUD patients, more severe ALD was comorbid condition (with a diagnosis 276 of alcoholic hepatitis). 

Also on page 9. It is not clear what is meant by the following statement - " thus we estimated if there is any mediating role of liver function/ injury in the hangover symptomology (Figure 5)."   If this was estimated, then how was it estimated?  Or was it hypothesised and then tested? 

On page 10,lines 292-293.  "This could lead to both the higher blood alcohol concentrations and their longer durations in the system."  The words the and their could be removed from the sentence.

Author Response

Thank you for allowing me to read your revised paper and addressing comments made previously.  I have some further comments if I may:

1) It is still unclear to me how the participants were recruited?

Response: We agree to further explain the recruitment effort for the study participants. We have added the details in the section 2.1.

2) While the small sample size used in the study has been acknowledged more directly now and discussed as a limitation, the study has also now been referred to  as a proof of principle study.  In my view, the 'exploratory'/ early stage nature of the study ought to now be reflected in the title and so the title of the paper ought to be amended.  In addition, there are still some statements in the paper that require to be re-worded, eg on p12 line 361/362 it is stated that " We acknowledge that expanding this study in a larger population would further support our results, and we are pursuing this presently".   However, given that it is not known that expanding the study would further support the results then a degree of caution ought to be exercised here such that the statement might say, for example, "We acknowledge that these initial results would benefit from further testing with a larger sample size, and we are pursuing this presently"

Response: we agree with the reviewer and have changed the wording at the specific location in the discussion section.

3) There are still a number of typographical errors/language issues evident in the paper that would benefit from re-checking, eg:

Response: We agree with the reviewer and have specifically made changes in the manuscript based on your following comments.

p2, lines 80/81 - Combined Neuroscience Institutional Review Board at the NIH, and the other approved by the University of Louisville IRB. IRBs of both the Institutions approved the study once concluded, meeting the ethical standards/regulations.

Response: We agree with the reviewer and have edited the statement to make it clear.

p3, line 98 - however, they were not abstinent (to alcohol drinking) and were actively drinking at the time of screening.

Response: We agree with the reviewer and have edited and simplified the statement.

Page 9. line 276 - there seems to be a word missing - In heavy drinking AUD patients, more severe ALD was comorbid condition (with a diagnosis 276 of alcoholic hepatitis).

Response: We agree with the reviewer and have edited the statement.

Also on page 9. It is not clear what is meant by the following statement - " thus we estimated if there is any mediating role of liver function/ injury in the hangover symptomology (Figure 5)."   If this was estimated, then how was it estimated?  Or was it hypothesised and then tested?

Response: We agree with the reviewer and replaced the word with a more appropriate verb in context.

On page 10,lines 292-293.  "This could lead to both the higher blood alcohol concentrations and their longer durations in the system."  The words the and their could be removed from the sentence.

Response: We agree with the reviewer and have changed the statement accordingly.

Reviewer 2 Report

The manuscript entitled “Exacerbation of Hangover Symptomology significantly corresponds with Heavy and Chronic Alcohol Drinking” aimed to identify how hangover symptoms contrast between moderate and heavy drinkers, to discover associations between hangover symptoms and heavy drinkers, and to analyze the interactions that modifiers may have on hangover symptoms and drinking patterns.

General: Grammar was reviewed throughout the paper. Minor grammatical errors were noted and detailed in this review.

Title: This article is appropriately named.

Abstract: In the opening sentence, it is stated that alcohol hangovers consist of different domains of symptoms. I suggest adding back the sympathetic domain in the opening sentence which has been removed, due to the sympathetic domain of symptoms being repeatedly referenced at later parts of the abstract and the rest of the paper. Upon my first read though of this abstract, I immediately noticed that the sample size seems questionable with a total of 41 people (12 in the heavy drinking group, 17 in the moderate drinking group, 12 in the social drinking group). This may lead other readers to question if there will be enough variability in this study to have valid results. I also do not notice any mention of the ethnicities of the patient population.

Introduction: In paragraph one, you mention that alcohol consumption is a leading cause of preventable mortality, and that alcohol consumption continues to steadily rise in the United States. Consider adding statistics to back up these claims if they exist. At the end of paragraph two, I would consider adding more symptoms listed for common hangover symptoms than the six that are present. Consider adding weakness, headaches, shakiness, excessive thirst, and dry mouth. Paragraph 3 does an excellent job of describing the lack of knowledge in this area and in establishing a need for this study. In the last sentence of paragraph four, consider removing liver disease as comorbid conditions would encompass this.

Study Participants and Methods: In section 2.1 titled “Patient recruitment,” add the word “of” before “the Institutions approved” and add the word “this” before “study once concluded.” When the authors say, “heavy drinkers drink around 15 drinks per day”, consider more explicitly stating that this is an average of this group if indeed it is. Continue this explicit statement of an average of the group for both the moderate and social drinking groups. The inclusion and exclusion criteria for the three groups are extensive and appropriate.

In section 2.2 titled “Study paradigm,” the authors do an adequate job of explaining the parameters that were collected to satisfy the timeline followback and Alcohol Hangover Scale criteria. However, some of the parameters may need clarification as to why they were chosen, if not part of guidelines, and the criteria that are part of guidelines should be specified versus those that are specific to this study. For example, a reader may wonder why the time of 10:00am was chosen as the cutoff time for reporting symptoms. The authors should specify if this time is part of guidelines, and, if not, why this time was chosen in this study.

In section 2.3 titled “Analysis,” the authors sufficiently explain their analytic methods. For liver function markers, a reader may wonder why gamma-glutamyl transferase (GGT), aspartate aminotransferase (AST) levels, and the ratio of AST levels versus alanine transaminase (ALT) levels were not mentioned here as this is classically used clinically to measure the severity of alcoholism. The finding of an AST to ALT ratio greater than 2:1, especially with elevated GGT, is more sensitive and much more specific for alcoholic liver disease. Although these liver injury markers are mentioned later in section 3.7, the authors should consider mentioning them here as well.

Results: In section 3.1 titled “Demographics and drinking profile,” the sample size of 41 was not discussed, and the effect this size would have on variability and the validity of the results. The authors should consider adding some discussion here about the sample size. The limitations of the sample size are most prominent in the exclusion of evaluation of sex differences due to only three female heavy drinkers. There was mention of very few females in the heavy drinking group, but there was no mention of almost double the number of females in the moderate group versus the males in the moderate group (11 females to 6 males). The authors should consider adding the breakdown of the ethnicities for these sample groups.

In section 3.2 titled “Assessment of Hangover symptoms,” in Table 2, shouldn’t the p-value be italicized for the hangover row since the p-value is >0.05 and therefore not a significant statistical difference? For Figures 1a through 1j, consider reformatting these as they are very small and hard to read the words on individual graphs. A reader looking at these same graphs cannot help but notice that the standard deviations are very large throughout all of the graphs. The small sample size could be causing this, will need to be pointed out here, and needs to be discussed at a later point.

In section 3.3 titled “AUDIT domains, and association of drinking markers in heavy drinkers,” the authors should consider reformatting these considered to make them larger and therefore easier to read. At minimum, an increase in the height of these figures should be considered. The sentence that was added saying, “suggesting overall AUDIT vs. heavy drinking relation as primarily due to the hazardous domain of AUDIT” may need to be removed from this section as it seems more appropriate for the discussion section. The authors should consider adding more figures of the dependence symptoms domain and harmful alcohol use domain versus heavy drinking days in the past 90 days for comparison to the statistically significant hazardous domain.

In section 3.4 titled “Internal consistency of hangover measures in heavy drinkers,” the authors did an adequate job explaining which symptoms of the domains showed statistical significance.

In section 3.5 titled “Association of AUDIT and hangover measures in heavy drinker,” the authors should once again consider changing the format to enlarge these figures; however, they did a sufficient job explaining the statistical associations between AUDIT and the different domains.

In section 3.6 titled “Role of AUDIT and heavy drinking TLFB markers on hangover symptoms,” the authors should consider showing the data for the “craving” measure and the AUDIT when adjusted for HDD90, as readers may be interested to see this data. The figures in this section are the easiest to read so far, however the height could still be increased some more. The authors should consider pointing out that the p-value for DS-AUDIT versus craving was 0.055 and therefore the only figure here that is not statistically significant.

In section 3.7 titled “Role of liver dysfunction on hangover symptoms,” the authors mention “we did not find any major effects of these liver injury markers in combination with AUDIT on hangover severity.” The authors should have considered acquiring AST, ALT, and GGT, showing, and plotting this data. Readers may want more information regarding this data, as this is the first values most clinicians will use to detect alcohol abuse. The authors should consider adding normal levels of the liver function and injury markers in table three. The authors should consider increasing the size of figure five to make it easier to read. I also question if the last paragraph belongs in the results section; it would seem more appropriate in the discussion.

Discussion: In the first sentence, consider replacing “Every heavy drinker” with “All heavy drinker subjects.” In paragraph six, the authors should expand upon the clinical significance of the association of liver function markers and higher hangover scale results. The authors should also mention that there was no major effect of liver injury markers (AST and ALT) with AUDIT on hangover severit in this study in this study.

For the limitations paragraph, the issue of the sample size was finally addressed as well as the lack of females in the heavy drinking group. I think the authors adequately addressed many of the limitations of the study, however, I would like to see the ethnical backgrounds also included in the demographics.

For the hangover symptoms that were experienced, additional data could have gathered as to the intensity of each one, and the length of time it was present. As with any where people answer questions, there is lots of subjectivity involved. Any method to include more objective data should be considered in future studies. Regarding the statistically significant results of this study, the clinical significance was never mentioned, therefore the authors should consider adding a paragraph addressing this.

Conclusion: This section is worded nicely. The authors adequately summarized the findings and limitations of this paper. In the last sentence, the authors should consider adding that larger population studies would also allow for analysis of differences between sexes.

Tables and Figures: Please consider reformatting the tables and figures throughout the paper to be larger in size for readers to better view them.

Overall: This paper has very few grammatical errors. The statistically significant results are clearly identified and explained. Increasing the size of the tables and figures, elaborating on the clinical significance, and adding normal ranges to table 3 would be beneficial.

Author Response

Review responses:

The manuscript entitled “Exacerbation of Hangover Symptomology significantly corresponds with Heavy and Chronic Alcohol Drinking” aimed to identify how hangover symptoms contrast between moderate and heavy drinkers, to discover associations between hangover symptoms and heavy drinkers, and to analyze the interactions that modifiers may have on hangover symptoms and drinking patterns.

General: Grammar was reviewed throughout the paper. Minor grammatical errors were noted and detailed in this review.

Response: We thank the reviewer to provide comments. We would edit the manuscript based on reviews received from both the reviewers.

Title: This article is appropriately named.

Abstract: In the opening sentence, it is stated that alcohol hangovers consist of different domains of symptoms. I suggest adding back the sympathetic domain in the opening sentence which has been removed, due to the sympathetic domain of symptoms being repeatedly referenced at later parts of the abstract and the rest of the paper. Upon my first read though of this abstract, I immediately noticed that the sample size seems questionable with a total of 41 people (12 in the heavy drinking group, 17 in the moderate drinking group, 12 in the social drinking group). This may lead other readers to question if there will be enough variability in this study to have valid results. I also do not notice any mention of the ethnicities of the patient population.

Response: We agree with the reviewer and have added the sympathetic domain back in the present revision draft.

We had received comments from two reviewers in the first round of review to explain the low number of participation in the study. We had provided the responses  when we submitted the revised version, which were satisfactory and accepted. Following is the response that we provided to each of the two reviewers during the 1st revision:

Reviewer 1: How was the sample size determined?

Response: This study is a proof of principle clinical investigation; thus, the study was performed on a moderate sized group of participants (to identify key outcomes and implement a large cohort study). In this study we have 41 subjects: 12 heavy drinkers, 17 moderate drinkers, and 12 social/light drinkers. 12 subjects per group enable us to detect an effect size (the ratio of group difference and the group standard deviation, assuming homogeneity of group variance) of 1.20 with power 80% at a significant level at 5%.  The majority of the primary endpoints in our study, which have statistically significant between-group differences, exhibit an effect size larger than 1.20. For example, the average hangover score of heavy drinkers and moderate drinkers is (3.3-1.14)/1.58=1.36. Thus, majority of the significant differences were obtained with power at least 80% at a significance level 0.05.

Our aim in the statistical design was to focus on the effect sizes. Thus, our study outcomes have mentioned the effects and narrowing their selection in the results section.

Reviewer 2: This study uses a very small sample size to investigate differences between moderate drinkers (n = 17) and very heavy drinkers (n = 12) in hangover symptoms. (1) The purpose was to prove something that is already known, since it has already been established that at a BAG of< .11 g% there is low reporting of hangover. (2) The very small sample size means the results are unstable and sample-bound to this small group; and means the analyses that include gender have ridiculously small cell sizes.  Therefore, this study does not add significant knowledge. However, the way the manuscript is typeset makes me think that the editor decided to accept it before it went out for reviews, so the rest of my comments are designed to improve the presentation.

Response: Our study participants who were heavy drinkers drank more than 14 drinks per drinking day. A BAG of .11 as mentioned by the reviewer, is 5-6 drinks at any given time and our heavy drinking group had a more than 2-fold of drinking profile. Our results showed that hangover symptomology with excessive drinking was considerably greater compared to moderate drinkers.

We agree with the reviewer that the study sample size is not large. This study was initiated based on some clinical observations, and thus a proof of principle study was needed and designed to study the drinking and hangover association in a heavy drinking population which shows uniquely excessive drinking  (average drinks per drinking day was almost 14 in comparison to 2.5 in moderate and 1.4 in social drinkers); thus sample sizes are low. We performed the sample size analysis prior to this investigation and the sample size is over the minimum needed to perform statistical analyses for the univariate analyses, n>6 in each group. However, the effect sizes are promising and are consistent with our previous observations. The intent of this study was to address the question and study the scope of the results. With these promising findings in a small set of participants as a pilot study, we have expanded our investigation in a larger population.

We have now provided ethnicity, most of the participants were Caucasian. Given the small size of the study participants, there is not much conclusions that can be drawn on race and ethnicity. We have added information on race and ethnicity in the results section. This issue has also been added as part of  limitation in the discussion section in the revised manuscript. 

Introduction: In paragraph one, you mention that alcohol consumption is a leading cause of preventable mortality, and that alcohol consumption continues to steadily rise in the United States. Consider adding statistics to back up these claims if they exist. At the end of paragraph two, I would consider adding more symptoms listed for common hangover symptoms than the six that are present. Consider adding weakness, headaches, shakiness, excessive thirst, and dry mouth. Paragraph 3 does an excellent job of describing the lack of knowledge in this area and in establishing a need for this study. In the last sentence of paragraph four, consider removing liver disease as comorbid conditions would encompass this.

Response: We agree with the reviewer and have mentioned the stats for alcohol consumption in the revised manuscript.

We have also added more symptoms of hangover in the relevant paragraph of the introduction. Lastly, we have removed liver disease from the introduction section as recommended by the reviewer.

Study Participants and Methods: In section 2.1 titled “Patient recruitment,” add the word “of” before “the Institutions approved” and add the word “this” before “study once concluded.” When the authors say, “heavy drinkers drink around 15 drinks per day”, consider more explicitly stating that this is an average of this group if indeed it is. Continue this explicit statement of an average of the group for both the moderate and social drinking groups. The inclusion and exclusion criteria for the three groups are extensive and appropriate.

Response: We agree with the reviewer and have made changes in section 2.1 (similar comments were provided by the second reviewer that we have accommodated in the same statement).

We have mentioned that the drinking frequency was “on an average”, wherever applicable.

In section 2.2 titled “Study paradigm,” the authors do an adequate job of explaining the parameters that were collected to satisfy the timeline followback and Alcohol Hangover Scale criteria. However, some of the parameters may need clarification as to why they were chosen, if not part of guidelines, and the criteria that are part of guidelines should be specified versus those that are specific to this study. For example, a reader may wonder why the time of 10:00am was chosen as the cutoff time for reporting symptoms. The authors should specify if this time is part of guidelines, and, if not, why this time was chosen in this study.

Response: We agree with the reviewer to clarify selection of timing of data collection.

We have mentioned about the guideline for selection of AHS information in the revised manuscript.

We have cited our previous publication’s paradigm (Vatsalya et al., 2018), in which we used the time of 10:00 am as cutoff for data collection.

In section 2.3 titled “Analysis,” the authors sufficiently explain their analytic methods. For liver function markers, a reader may wonder why gamma-glutamyl transferase (GGT), aspartate aminotransferase (AST) levels, and the ratio of AST levels versus alanine transaminase (ALT) levels were not mentioned here as this is classically used clinically to measure the severity of alcoholism. The finding of an AST to ALT ratio greater than 2:1, especially with elevated GGT, is more sensitive and much more specific for alcoholic liver disease. Although these liver injury markers are mentioned later in section 3.7, the authors should consider mentioning them here as well.

Response: We agree with the reviewer to elaborate on the liver injury markers in the methods section.

We have added the liver injury markers that we have evaluated in this investigation (methods) in the revised manuscript.

We did not examine GGT in our study and have mentioned as a limitation in the discussion section.

Results: In section 3.1 titled “Demographics and drinking profile,” the sample size of 41 was not discussed, and the effect this size would have on variability and the validity of the results. The authors should consider adding some discussion here about the sample size. The limitations of the sample size are most prominent in the exclusion of evaluation of sex differences due to only three female heavy drinkers. There was mention of very few females in the heavy drinking group, but there was no mention of almost double the number of females in the moderate group versus the males in the moderate group (11 females to 6 males). The authors should consider adding the breakdown of the ethnicities for these sample groups.

Response: We agree with the reviewer that the sample size was not discussed in the section 3.1. We have mentioned the sample size in the section 2.1; and have also been presented this information in the Table 1. This information would be redundant in the introductory subsection of the Results section.

We have discussed about the limited participants in the discussion section, where other limitations are mentioned.

We have mentioned the number of female participants in the moderate drinkers in the revised manuscript.

Regarding the race/ethnicity data, we have briefly mentioned in the section 3.1 of the Results section in the revised manuscript. There were very few individuals who were not Caucasian.

In section 3.2 titled “Assessment of Hangover symptoms,” in Table 2, shouldn’t the p-value be italicized for the hangover row since the p-value is >0.05 and therefore not a significant statistical difference? For Figures 1a through 1j, consider reformatting these as they are very small and hard to read the words on individual graphs. A reader looking at these same graphs cannot help but notice that the standard deviations are very large throughout all of the graphs. The small sample size could be causing this, will need to be pointed out here, and needs to be discussed at a later point.

Response: We agree with the reviewer and have italicized the p-values that are not significant.

We have attempted to enlarge the sub-figures of the figure 1 to make the words bigger. We have noted about the large standard deviations in this section.

In section 3.3 titled “AUDIT domains, and association of drinking markers in heavy drinkers,” the authors should consider reformatting these considered to make them larger and therefore easier to read. At minimum, an increase in the height of these figures should be considered. The sentence that was added saying, “suggesting overall AUDIT vs. heavy drinking relation as primarily due to the hazardous domain of AUDIT” may need to be removed from this section as it seems more appropriate for the discussion section. The authors should consider adding more figures of the dependence symptoms domain and harmful alcohol use domain versus heavy drinking days in the past 90 days for comparison to the statistically significant hazardous domain.

Response: We agree with the reviewer and have enlarged the figure. HDD90 and other drinking markers did not show any significant associations with AUDIT (or with the other domains of AUDIT), therefore we have not plotted them. It shows that AUDIT showed direct relationship with HDD90, and the single domain that corresponding was important with HDD90 was hazardous domain. To avoid plotting any non-significant, we added the statement “suggesting overall AUDIT vs. heavy drinking relation as primarily due to the hazardous domain of AUDIT”. 

Discussion of this association is already present in the first paragraph of the Discussion section.

In section 3.4 titled “Internal consistency of hangover measures in heavy drinkers,” the authors did an adequate job explaining which symptoms of the domains showed statistical significance.

Response: We thank the reviewer for noting this information.

In section 3.5 titled “Association of AUDIT and hangover measures in heavy drinker,” the authors should once again consider changing the format to enlarge these figures; however, they did a sufficient job explaining the statistical associations between AUDIT and the different domains.

Response: we agree with the reviewer and have enlarged the figure.

In section 3.6 titled “Role of AUDIT and heavy drinking TLFB markers on hangover symptoms,” the authors should consider showing the data for the “craving” measure and the AUDIT when adjusted for HDD90, as readers may be interested to see this data. The figures in this section are the easiest to read so far, however the height could still be increased some more. The authors should consider pointing out that the p-value for DS-AUDIT versus craving was 0.055 and therefore the only figure here that is not statistically significant.

Response: We agree with the reviewer and have rerun this specific stat. Reviewer provided a very valuable insight. We found an even stronger effect of this association. We have added this information in the section 3.6. We have also mentioned the about the insignificant p-value of Fig. 4c in the revised manuscript.

We have further enlarged the Fig. 4.

In section 3.7 titled “Role of liver dysfunction on hangover symptoms,” the authors mention “we did not find any major effects of these liver injury markers in combination with AUDIT on hangover severity.” The authors should have considered acquiring AST, ALT, and GGT, showing, and plotting this data. Readers may want more information regarding this data, as this is the first values most clinicians will use to detect alcohol abuse. The authors should consider adding normal levels of the liver function and injury markers in table three. The authors should consider increasing the size of figure five to make it easier to read. I also question if the last paragraph belongs in the results section; it would seem more appropriate in the discussion.

Response: We agree with the reviewer that ALT, AST and GGT are important markers of liver injury. We conducted analyses for AST and ALT. We did not find similar interaction as we found with albumin and total bilirubin (which are indicator of liver function). On the reasons could be that albumin and total bilirubin show liver function status, whereas ALT and AST are indicators of liver injury. Thus, our hypothesis that deficient liver function was better predictor of changes in alcohol metabolism than the liver injury markers that could have caused higher hangover symptoms. We have added more clinical inferences in the discussion section in the revised manuscript. We did not analyze GGT in these patients and have noted as a limitation of the study.

Discussion: In the first sentence, consider replacing “Every heavy drinker” with “All heavy drinker subjects.” In paragraph six, the authors should expand upon the clinical significance of the association of liver function markers and higher hangover scale results. The authors should also mention that there was no major effect of liver injury markers (AST and ALT) with AUDIT on hangover severit in this study in this study.

Response: We agree with the reviewer and have made the changes in the revised manuscript. We have expanded the association of liver function and hangover in the discussion section. We have also noted about the lack of effects of liver injury in comparison to the liver function status, contributing to higher hangover scores.

For the limitations paragraph, the issue of the sample size was finally addressed as well as the lack of females in the heavy drinking group. I think the authors adequately addressed many of the limitations of the study, however, I would like to see the ethnical backgrounds also included in the demographics.

Response: We appreciate reviewer’s observation for our detailed limitation of this study. We have provided the ethnical background of the subjects as much as possible in the results section of the revised manuscript.

For the hangover symptoms that were experienced, additional data could have gathered as to the intensity of each one, and the length of time it was present. As with any where people answer questions, there is lots of subjectivity involved. Any method to include more objective data should be considered in future studies. Regarding the statistically significant results of this study, the clinical significance was never mentioned, therefore the authors should consider adding a paragraph addressing this.

Response: We agree with the reviewer for both the comments. For the first stipulation, we have added a relevant statement in the limitation paragraph of the discussion section.

We have further added the clinical relevance of this study in the discussion section of the revised manuscript.

Conclusion: This section is worded nicely. The authors adequately summarized the findings and limitations of this paper. In the last sentence, the authors should consider adding that larger population studies would also allow for analysis of differences between sexes.

Response: We agree with the reviewer and have reworded the last sentence of the conclusion.

Tables and Figures: Please consider reformatting the tables and figures throughout the paper to be larger in size for readers to better view them.

Response: we agree with the reviewer. We have improved the tables. Figures are Tiff files, and they can be expanded by the publication manager per the needs if the manuscript is approved for publication.

Overall: This paper has very few grammatical errors. The statistically significant results are clearly identified and explained. Increasing the size of the tables and figures, elaborating on the clinical significance, and adding normal ranges to table 3 would be beneficial.

Response: We thank the reviewer for the close observation. The page width limited the presentation of the figures and tables. If manuscript accepted, we will work with the publication manager closely to emphasize the reviewer’s suggestions to present the figures and tables in the best way possible.